

# The frequency of combined IFITM3 haplotype involving the reference alleles of both rs12252 and rs34481144 is in line with COVID-19 standardized mortality ratio of ethnic groups in England

Dimitris Nikoloudis, Dimitrios Kountouras and Asimina Hiona

Center for Preventive Medicine & Longevity, Bioiatriki Healthcare Group, Athens, Attiki, Greece

## ABSTRACT

Evidence was brought forward in England and the USA that Black, Asian, Latino and Minority Ethnic people exhibit higher mortality risk from COVID-19 than White people. While socioeconomic factors were suggested to contribute to this trend, they arguably do not explain the range of the differences observed, allowing for possible genetic implications. Almost concurrently, the analysis of a cohort in Chinese COVID-19 patients proposed an association between the severity of the disease and the presence of the minor allele of rs12252 of the Interferon-induced transmembrane protein 3 (IFITM3) gene. This SNP, together with rs34481144, are the two most studied polymorphisms of IFITM3 and have been associated in the past with increased severity in Influenza, Dengue, Ebola, and HIV viruses. IFITM3 is an immune effector protein that is pivotal for the restriction of viral replication, but also for the regulation of cytokine production. Following up on these two developments in the ongoing SARS-CoV-2 pandemic, the present study investigates a possible association between the differences in mortality of ethnic groups in England and the combined haplotypes of rs12252 and rs34481144. The respective allele frequencies were collected for 26 populations from the 1000 Genomes Project and subgroups were pooled wherever possible to create correspondences with ethnic groups in England. A significant correlation ($r = 0.9687$, $p = 0.0003$) and a striking agreement was observed between the reported Standardized Mortality Ratios and the frequency of the combined haplotype of both reference alleles, suggesting that the combination of the reference alleles of the specific SNPs may be implicated in more severe outcomes of COVID-19. This study calls for further focus on the role of IFITM3 variants in the mechanism of cellular invasion of SARS-CoV-2, their impact in COVID-19 severity and their possible implications in vaccination efficacy.

## INTRODUCTION

Emerging scientific evidence from international (*Kirby, 2020*) and UK (*Aldridge et al., 2020*) COVID-19 patient reports and death records, indicate a disproportionate effect of

Corresponding author
Dimitris Nikoloudis,
dnicolgr@hotmail.com

the novel coronavirus on ethnic minorities. According to CDC (*Centers for Disease Control and Prevention, 2020*), Black, Asian and Minority Ethnic (BAME) people are at higher risk of death from COVID-19. Importantly, an Indirect Standardization of NHS mortality data in England (*Aldridge et al., 2020*), revealed that the adjusted for age and region Standardized Mortality Ratios (SMRs), were highest in Black African, Black Caribbean, Pakistani, Bangladeshi, and Indian minority ethnic groups. In contrast, White Irish and White British ethnic groups exhibited a significantly lower risk of death. Similarly, in the USA (*Garg et al., 2020*), preliminary data compiled from hospitals in 14 US states, confirmed the UK study outcomes, showing that African Americans are also disproportionately affected by COVID-19. Specifically, African Americans represented 33% of COVID-19 hospitalizations, despite only making up 18% of the total population studied. In a subsequent analysis, among COVID-19 deaths in New York City, for which race and ethnicity data were available, death rates from COVID-19 among black or African Americans and Hispanic or Latinos were substantially higher than that of White or Asian people (*Garg et al., 2020*).

Several reasons have been proposed to explain these ethnic discrepancies in COVID-19 mortality risk arising from these preliminary studies. Chronic pre-existing conditions, such as cardiovascular diseases (CVD), diabetes, hypertension, obesity, etc. are more common in minorities compared to Caucasian populations and have all been associated with adverse outcomes in COVID-19 (*Centers for Disease Control and Prevention, 2020*; *Kirby, 2020*). However, race disparities in those diseases are not large enough to fully explain the COVID-19 death disparity (*Aldridge et al., 2020*). Factors such as housing and living conditions, use of public transportation, lack of regular access to primary health, and occupation-related differences that prohibit the work from home, or require more frequent and/or close social contact, may have all played an important role in producing disproportionate death rates among BAME groups (*Aldridge et al., 2020*; *Kirby, 2020*; *Niedzwiedz et al., 2020*; *Khunti et al., 2020*). Nevertheless, it is suggested that inequalities in socioeconomic status parameters do not seem to adequately explain the range of differences, and in some instances, the extreme variations observed among ethnic minorities in mortality rates from COVID-19 infection (*Kirby, 2020*).

As the importance of genetic polymorphisms (SNPs) in the modulation of individual susceptibility to, and severity of, infectious diseases has been well established (*Chapman & Hill, 2012*; *Zhao et al., 2018*), we turned our focus to two very highly studied polymorphisms of the interferon-induced transmembrane protein 3 (IFITM3) gene: rs12252 and rs34481144. IFITM3 encodes an immune effector protein that is pivotal for restriction of viral replication (*Brass et al., 2009*) of many enveloped RNA viruses including HIV-1, influenza A virus (IAV), Ebola and Dengue virus (*Brass et al., 2009*; *Feeley et al., 2011*; *Huang et al., 2011*; *Everitt et al., 2012*; *Compton et al., 2014*). IFITM3 has also been demonstrated to affect severity of infection and improve the host cellular defenses against viruses (*Brass et al., 2009*; *Everitt et al., 2012*; *Compton et al., 2014*). Interestingly, IFITM3 has also been shown to act as a regulator of antiviral immunity that controls cytokine production to restrict viral pathogenesis, in CMV (*Stacey et al., 2017*) and Sendai virus (*Jiang et al., 2017*). This finding is particularly important since cytokine storm in

influenza can lead to a rapid progression of the infection in humans (*Wang et al., 2014*) and the same observation is also apparent in COVID-19 severe and deadly cases (*Giamarellos-Bourboulis et al., 2020*; *Blanco-Melo et al., 2020*). Moreover, IFITM3 was found to be explicitly upregulated in SARS-CoV-2 infected cells (*Blanco-Melo et al., 2020*; *Hachim et al., 2020*; *He et al., 2020*).

The minor allele of rs12252 (C in minus, or G in plus strand orientation) has been associated with rapid progression of acute HIV infection (*Zhang et al., 2015*), with the severity of influenza (*Zhang et al., 2013*) and recently with COVID-19 severity (*Zhang et al., 2020*). The minor allele of rs34481144 (A in minus, or T in plus strand orientation) was previously found to be correlated with increased severity of IAV infection (*Allen et al., 2017*). Moreover, the minor allele of rs34481144 is also associated with enhanced methylation on the IFITM3 promoter of CD8+ T cells, and general transcriptional repression of the broader locus surrounding IFITM3, which includes several genes known to be involved in host responses to viral infection (*Wellington et al., 2019*).

SARS-CoV-2 uses primarily the ACE2 receptor as main point of entry and the host cell serine protease TMPRSS2 for viral spike (S) protein priming (*Hoffmann et al., 2020*). Severe acute respiratory syndrome coronavirus (SARS-CoV), which also uses ACE2 as a receptor, has been shown to be restricted more efficiently by IFITM1 than by IFITM3, presenting a different restriction pattern than IAV (*Huang et al., 2011*). Interestingly, it was recently shown that TMPRSS2 is specifically allowing evasion of IFITM3 restriction for bat SARS-Like WIV1 coronavirus (*Zheng et al., 2020*), opening the possibility for a similar mechanism in the case of SARS-CoV-2. Further potential involvement of IFITM3 in COVID-19 outcome was revealed in the context of syncytial pneumocytes in severe cases with extensive lung damage, where it was suggested that the cellular location of IFITMs 1–3 could be playing a role in syncytia formation (*Buchrieser et al., 2020*). Indeed, the accumulation of many direct and indirect layers of evidence linking IFITM3 with COVID-19 severity, has also led to explicit calls for further investigation of the role of this highly relevant first-line of cellular defense protein (*Zhao, 2020*). Following up to the analysis of COVID-19 NHS mortality data in BAME groups (*Aldridge et al., 2020*), the purpose of the present study was to investigate a possible association between the stand-alone and combined frequencies of the alleles of the IFITM3 gene variants rs12252 and rs34481144, with COVID-19 standardized mortality ratio of ethnic groups in England.

The sole incentive of the current research is to help improve the existing and future treatment protocols for severe COVID-19 patients, and in no case to provide DNA-based arguments that may be used to mask existing social inequalities or racism.

## METHODS

The Standardized Mortality Ratios (SMR) of ethnic groups in England, adjusted for age and region, were adopted from the study by *Aldridge et al. (2020)* in the exact form in which they were presented. Specific dataset details, including age, region and ethnicity information, are available at the repository address of the respective publication (https://discovery.ucl.ac.uk/id/eprint/10096589). The rs12252 (A>G) and rs34481144 (C>T) allele and haplotype frequencies were collected for all available 1000 Genomes

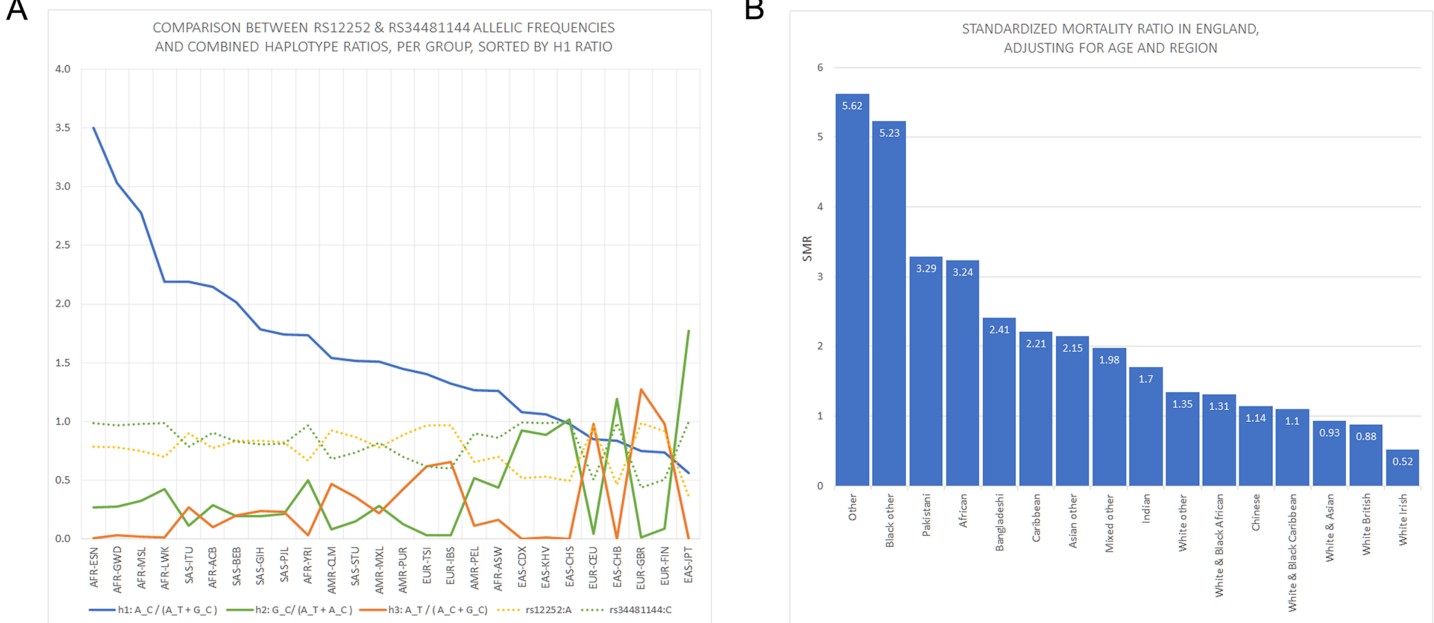

**Figure 1** **Panels showing rs12252 and rs34481144 allelic frequencies vs Standardized Mortality Ratios in England.** (A) Comparison of h1 [A_C/(A_C+G_C): blue line], h2 [G_C/(A_T+A_C): green line] and h3 [A_T/(A_C+G_C): orange line] haplotype ratios, for all available ethnic subgroups in the 1000 Genomes Project. The fluctuation of frequencies of the major alleles rs12252:A (yellow dotted line) and rs34481144:C (green dotted line) follows a trend similar to h3 and h2 ratios, respectively. On the contrary, h1 ratio shows a unique trend. (B) Standardized Mortality Ratio (SMR) of ethnic groups in England, adjusted for Age and NHS Region.           

Project ancestral populations, from LDlink (LDhap tool: https://ldlink.nci.nih.gov/?tab=ldhap), specifically five major groups, that is, African (AFR), Ad Mixed American (AMR), European (EUR), East Asian (EAS) and South Asian (SAS), comprising 26 subgroups in total (*Machiela & Chanock, 2015*) (Table 1). The plus orientation for the reference and minor alleles was retained throughout this analysis, for better data handling and in compliance with dbSNP. Combined rs12252_rs34481144 haplotypes were defined as A_C (H1), G_C (H2), A_T (H3), while G_T haplotype was not represented at all in the data Rankings were examined by sorting all populations by individual reference allele (rs12252:A, rs34481144:C) and by combined haplotype frequency ratios (rs12252_rs34481144: h1_ratio=A_C/(A_T+G_C), h2_ratio=G_C/(A_C+A_T), h3_ratio=A_T/(G_C+A_C)) (Fig. 1A), and subsequently compared visually to the reported Standardized Mortality Ratios (SMR) of ethnic groups in England (Figs. 1B and 2). Additionally, an attempt was made to correlate directly the two rankings, that is, SMR and IFITM3 haplotype frequencies by specific reported ethnic subgroup. UK demographics sources were therefore consulted (*Office for National Statistics, UK, 2011*; *Chanda & Ghosh, 2012*) in order to pool, wherever possible, the ancestral subgroups to the reported ethnic groups in England. With all reservations tied to the inevitable discrepancies of this type of simplified socio-genetic correspondences, the following pools were formed: [AFR-YRI, AFR-LWK, AFR-GWD, AFR-MSL, AFR-ESN]>"African", [SAS-STU, SAS-GIH, SAS-PJL]>"Indian", [EAS-CDX, EAS-CHS, EAS-CHB]>"Chinese", [EUR-CEU, EUR-IBS, EUR-TSI, EUR-FIN]>"White Other" (see Table 1 for full subgroup

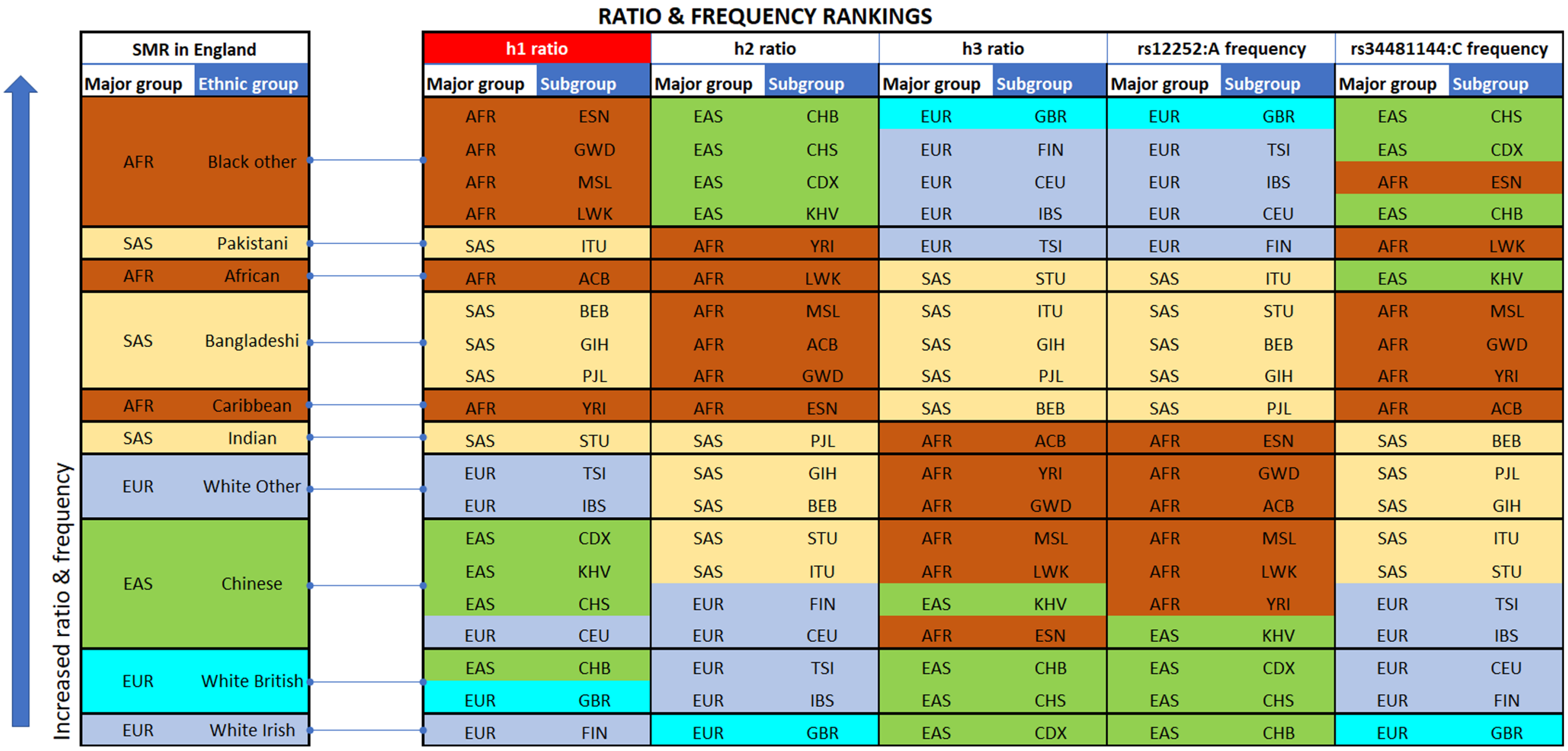

**Figure 2 Vertical stacking showing increasing rankings of SMR in England, rs12252_ rs34481144 haplotype ratios and allele frequencies for the various Major groups and Subgroups (highest on top).** East Asian subgroups (in green) present the highest ranking among all 1000 Genomes Project populations in h2 ratios and rs34481144:C frequencies (5th column from left and second to last column), while European subgroups (in blue) present the highest ranking in h3 ratios and rs12252:A frequencies (7th & 9th column from left). H1 haplotype ratio ranking (3rd column from left) presents an almost identical alignment with the reported SMR of major groups in England (1st column from left). Note: ranked items are color-tagged by their major group, that is, continent of origin: AFR, SAS, EAS, EUR—no subgroup pooling is shown here (see Table 2 for pooling details).

descriptions and Table 2 for pooled subgroups and correspondences to ethnic minorities). A pool for the reported Pakistani group failed to form from ancestral populations, as the Punjabi (SAS-PJL), being the only related subgroup, account roughly for just 45% of Pakistan's demographics, while in London the community includes comparable numbers of Punjabis, Pathans and Kashmiris, with small communities of Sindhis and Balochis (*Department for Communities & Local Government, UK, 2009*). Moreover, the Punjabi form also a considerable part of Indians' pool (at least 40% of Delhi's total population), therefore a single-ended direct correspondence between Punjabi and British Pakistani was not warranted in this case. Indian Telugu (SAS-ITU) were not included in Indians' pool, as no demographic report was suggestive of comparable numbers to the other 3 included subgroups, for people of Indian origin in England. The same rationale applied for the non-inclusion of AFR-ASW (Americans of African Ancestry in SW USA) in the African pool. The haplotype frequencies were simply averaged within pooled groups, and both the ratios and SMR were normalized to the White British result (represented uniquely by EUR-GBR subgroup) (Table 2).

# RESULTS

We extracted rs12252 and rs34481144 allele frequencies of various ethnic groups from the 1000 Genomes Project, in order to examine whether the distribution of any one of the

**Table 1 Detailed allele and haplotype frequencies per ethnic subgroup derived from the 1000 Genomes Project for rs12252 and rs34481144.**

| Major group | Subgroup | rs12252:A | rs12252:G | rs34481144:C | rs34481144:T | A_T | A_C | G_C |
|---|---|---|---|---|---|---|---|---|
| EUR | CEU (Utah residents from north and west Europe) | 0.955 | 0.045 | 0.505 | 0.495 | 0.49 | 0.46 | 0.05 |
| EUR | TSI (Toscani in Italia) | 0.967 | 0.033 | 0.617 | 0.383 | 0.38 | 0.58 | 0.03 |
| EUR | FIN (Finnish in Finland) | 0.919 | 0.081 | 0.505 | 0.495 | 0.49 | 0.42 | 0.08 |
| EUR | GBR (British in England & Scotland) | 0.989 | 0.011 | 0.440 | 0.560 | 0.56 | 0.43 | 0.01 |
| EUR | IBS (Iberian Population in Spain) | 0.967 | 0.033 | 0.603 | 0.397 | 0.40 | 0.57 | 0.03 |
| EAS | CHB (Han Chinese in Beijing) | 0.461 | 0.539 | 0.990 | 0.010 | 0.00 | 0.45 | 0.54 |
| EAS | JPT (Japanese in Tokyo) | 0.361 | 0.639 | 1.000 | 0.000 | 0.00 | 0.36 | 0.64 |
| EAS | CHS (Southern Han Chinese) | 0.495 | 0.505 | 1.000 | 0.000 | 0.00 | 0.50 | 0.50 |
| EAS | CDX (Chinese Dai in Xishuangbanna | 0.521 | 0.478 | 0.995 | 0.005 | 0.00 | 0.52 | 0.48 |
| EAS | KHV (Kinh in Ho Tsi Minh ciry, Vietnam) | 0.530 | 0.470 | 0.985 | 0.015 | 0.02 | 0.52 | 0.47 |
| AMR | MXL (Mexican ancestry from Los Angeles) | 0.781 | 0.219 | 0.820 | 0.180 | 0.18 | 0.60 | 0.22 |
| AMR | PUR (Puerto Ricans from Puerto Rico) | 0.889 | 0.111 | 0.702 | 0.298 | 0.30 | 0.59 | 0.11 |
| AMR | CLM (Colombians from Medelin, Colombia) | 0.925 | 0.074 | 0.681 | 0.319 | 0.32 | 0.61 | 0.07 |
| AMR | PEL (Peruvians from Lima Peru) | 0.659 | 0.341 | 0.900 | 0.100 | 0.10 | 0.56 | 0.34 |
| SAS | GIH (Gujarati Indian from Houston Texas) | 0.835 | 0.165 | 0.806 | 0.194 | 0.19 | 0.64 | 0.17 |
| SAS | PJL (Punjabi from Lahore, Pakistan) | 0.823 | 0.177 | 0.812 | 0.188 | 0.19 | 0.64 | 0.18 |
| SAS | BEB (Bengali from Bangladesh) | 0.837 | 0.163 | 0.831 | 0.169 | 0.17 | 0.67 | 0.16 |
| SAS | STU (Sri Lankan Tamil from the UK) | 0.868 | 0.132 | 0.735 | 0.265 | 0.26 | 0.60 | 0.13 |
| SAS | ITU (Indian Telugu from the UK) | 0.897 | 0.103 | 0.789 | 0.211 | 0.21 | 0.69 | 0.10 |
| AFR | ASW (Americans of African Ancestry in SW USA) | 0.697 | 0.303 | 0.861 | 0.139 | 0.14 | 0.56 | 0.30 |
| AFR | ACB (African Carribeans in Barbados) | 0.776 | 0.224 | 0.906 | 0.094 | 0.09 | 0.68 | 0.22 |
| AFR | ESN (Esan in Nigera) | 0.788 | 0.212 | 0.990 | 0.010 | 0.01 | 0.78 | 0.21 |
| AFR | MSL (Mende in Sierra Leone) | 0.753 | 0.247 | 0.982 | 0.018 | 0.02 | 0.74 | 0.25 |
| AFR | GWD (Gambian in Western Gambia) | 0.783 | 0.217 | 0.969 | 0.031 | 0.03 | 0.75 | 0.22 |
| AFR | LWK (Luhya in Webuye Kenya) | 0.702 | 0.298 | 0.985 | 0.015 | 0.02 | 0.69 | 0.30 |
| AFR | YRI (Yoruba in Ibadan, Nigera) | 0.667 | 0.333 | 0.968 | 0.032 | 0.03 | 0.63 | 0.33 |

combined haplotypes is directly correlated with the reported SARS-CoV-2 related SMR in England. At first we compared the trend lines of reference allele frequencies with those of combined haplotype ratios (Fig. 1A). The fluctuation of frequencies of major alleles rs12252:A and rs34481144:C follows a trend similar to h3 and h2 ratios, respectively, while h1 ratio shows a unique trend. The ranking that visually appeared in line with the reported SMR, adjusted for age and NHS region (Fig. 1B), was produced by the h1 ratio. Two levels of SMR vs haplotype comparisons were applied: first, we compared at the level of major groups (i.e., un-pooled comparison: EAS vs SAS vs EUR vs AFR, see Fig. 2), second, at the level of ethnic subgroups, wherever possible (pooled comparison, see Table 2). For the evaluation of un-pooled rank correlation we considered two scenarios. First, we considered all 10 ethnic groups with an available SMR (Fig. 2) and we observed an almost perfect alignment with h1 ratios, which corresponds to a permutation event of 10 items, with 1/3,628,800 chance to occur randomly, corresponding to $p = 3 \times 10^{-7}$ (5σ). Secondly, we considered only 5 aligned items, specifically the sequence of (a) African
**Table 2 Pools of ethnic subgroups from the 1000 Genomes Project were formed to emulate the ethnic populations that are reported in England.**

| Ethnic group | SMR-White British Normalized | h1 ratio A_C/(A_T+G_C)—White British Normalized | 1000 Genomes Populations | rs12252:A | rs12252:G | rs34481144:C | rs34481144:T | A_T | A_C | G_C | rs12252_rs34481144 : h1 ratio A_C/(A_T+G_C) |
|---|---|---|---|---|---|---|---|---|---|---|---|
| African | 3.68 | 3.38 | AFR-YRI, AFR-LWK, AFR-GWD, AFR-MSL, AFR-ESN | 0.74 | 0.26 | 0.98 | 0.02 | 0.02 | 0.72 | 0.26 | 2.54 |
| Bangladeshi | 2.74 | 2.69 | SAS-BEB | 0.84 | 0.16 | 0.83 | 0.17 | 0.17 | 0.67 | 0.16 | 2.02 |
| Caribbean | 2.51 | 2.86 | AFR-ACB | 0.78 | 0.22 | 0.91 | 0.09 | 0.09 | 0.68 | 0.22 | 2.15 |
| Indian | 1.93 | 2.23 | SAS-STU, SAS-GIH, SAS-PJL | 0.84 | 0.16 | 0.78 | 0.22 | 0.22 | 0.63 | 0.16 | 1.68 |
| White other | 1.53 | 1.38 | EUR-CEU, EUR-IBS, EUR-TSI, EUR-FIN | 0.95 | 0.05 | 0.56 | 0.44 | 0.44 | 0.51 | 0.05 | 1.04 |
| Chinese | 1.30 | 1.28 | EAS-CDX, EAS-CHS, EAS-CHB | 0.49 | 0.51 | 1.00 | 0.01 | 0.00 | 0.49 | 0.51 | 0.96 |
| White British | 1.00 | 1.00 | EUR-GBR | 0.99 | 0.01 | 0.44 | 0.56 | 0.56 | 0.43 | 0.01 | 0.75 |
| Average | 2.10 | 2.12 | | 0.80 | 0.20 | 0.78 | 0.22 | 0.21 | 0.59 | 0.20 | 1.59 |

**Note:**
Column 1 shows Ethnic groups in England, while column 4 shows the selected pooling for each Ethnic group. Allele (columns 5–8 from left) and haplotype (columns 9–11 from left) frequencies, as well as h1 ratio (last column) were re-calculated for each pool. Both the h1 ratio [A_C/(A_T+G_C)] and the Standardized Mortality Ratios (SMR) were normalized by the corresponding numbers of White British (columns 2 and 3 from left), to allow for a direct comparison.

groups, followed by (b) South Asian, followed by (c) White Non-British, (d) Chinese, and finally (e) White British, a permutation event with 1/120 chance to occur randomly, or $p = 0.008$. In both cases the correlations proved highly significant. It is noteworthy, that subgroups EUR-IBS (Iberian Population in Spain) and EUR-TSI (Toscani in Italia), representative of two countries that suffered higher death rates than other European countries, share the highest h1 ratio between all European subgroups. Subsequently, in order to assess the potential strength of the theorized association, the rankings of pooled h1 ratios and SMR, per group, were linearly and significantly correlated, with Pearson $r = 0.9687$, $p = 3 \times 10^{-4}$ (>3.5σ) (Fig. 3).

## DISCUSSION

The calculated level of correlation appears to be remarkable, considering the possible discrepancies in the pooling of the available ancestral groups, but also the expected multi-parametric causes of the observed COVID-19 SMR in England's ethnic groups (as previously described, potentially involving prior health status, income level, household density, behavioral biases, questionable attribution of death to COVID-19, etc.). On one hand, the alignment of ethnic group rankings between SMR and un-pooled h1 ratio, is less than 1/120 probable to occur randomly at the worst case, and less than 1 in 3.6 million probable to occur randomly at the best case. In other words, if one hypothesizes that the reported SMR rankings are solely due to socioeconomic factors, then one would conclude that socioeconomic factors would be in perfect alignment with h1 ratios. The possibility of the above to occur seems highly unlikely, thus pointing to the fact that differential allele frequencies play a potentially important role in the reported SMR.
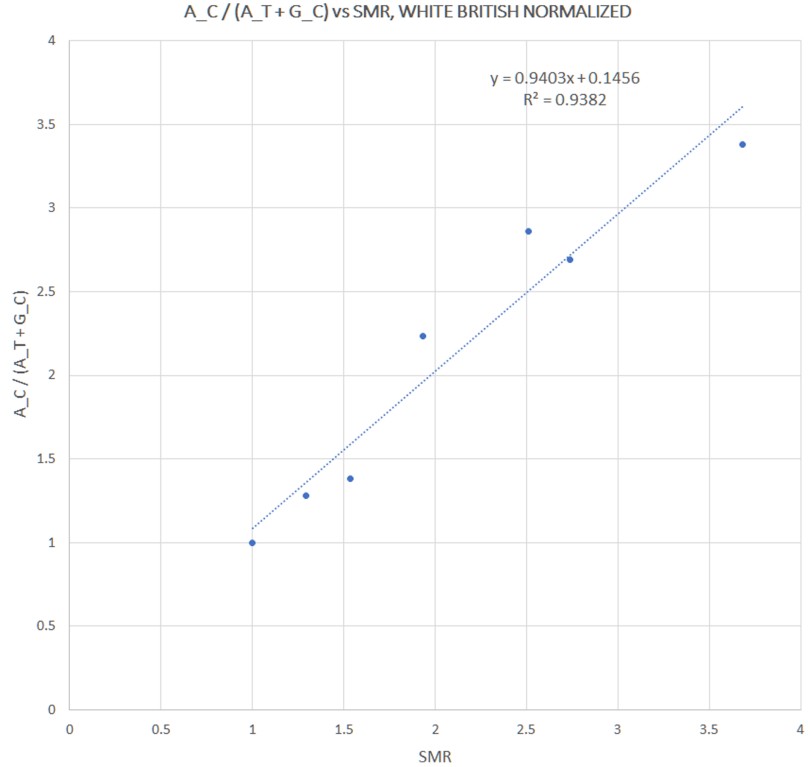

**Figure 3 Correlation between pooled h1 haplotype ratios and Standardized Mortality Ratios (SMR), with Pearson $r = 0.9687$, $p = 3 \times 10^{-4}$.**

On the other hand, the level of correlation ($>3.5\sigma$) between SMR and pooled h1 ratio, confirms the previous alignment and appears strong enough to suggest a possible causal link, albeit in this case, the pooling process may have introduced some discrepancies. The potential introduction of pooling discrepancies is expected and is impossible to quantify given the available data. However, the purpose of the pooled analysis was mainly to reinforce the primary association between IFITM3 and COVID-19 severity rather than to (indirectly) infer causality. Considering the parametric uncertainty of the pooled analysis it is probably inadequate to suggest a causal link, based on the strength of the observed correlation alone. However, taken together, this set of results constitutes a clear and valid starting point for designing further investigations regarding the role of IFITM3 in COVID-19 severity and appears as one more piece of evidence towards this direction. Therefore, the main point of this analysis is that the two examined SNPs should preferably henceforth be studied under a combined haplotype and not separately, as was performed so far for SARS-CoV-2, and a great variety of other viruses.

Before proceeding to the discussion of the implications of the above conclusion, it is interesting to also view the present observations in the context of data from the USA. A preliminary analysis of death rates from COVID-19 in New York City shows 92.3 deaths per 100,000 population among black or African American people, followed by Hispanic or Latino people (74.3), then by white (45.2) or Asian (34.5) people (*Kirby, 2020*). The same trend was clearly displayed in the initial ranking by h1 ratio (Fig. 1A), with

American populations (AMR) occupying the middle of the chart, between African and European/Asian populations. Interestingly, the reported lower death rate of Asian people compared to white people in the New York data, which represents an inversion of the respective numbers in England (Fig. 1B), could be justified by the fact that White British likely constitute a smaller proportion of the reported "white phenotype" in the USA (1.5 M in USA & Canada) and likewise, Japanese people possibly constitute a bigger proportion of of the reported "Asian phenotype" in the USA (>1.5 M). Both White British (EUR-GBR) and Japanese (EAS-JPT) ethnic subgroups have among the lowest h1 ratio between all subgroups.

The fact that the proposed risk haplotype (A_C) involves the reference alleles of both studied SNPs, appears as counterintuitive. Especially so, since it has been suggested after analysis of a Chinese cohort, that it is the minor allele rs12252:G that is linked to increased COVID-19 severity (Zhang et al., 2020). In fact the minor allele rs12252:G was linked to worse outcome in almost every related study, such as increased influenza severity (Zhang et al., 2013), or more rapid HIV progression (Zhang et al., 2015), although always observed in Chinese patients and not in European or American cohorts. This is noteworthy, as minor allele rs12252:G is found frequently in Chinese populations (roughly 50%), but is on the contrary rare in European populations (1–8%), or infrequent in South Asian (10–18%), or African groups (21–33%). Interestingly, an inverted trend is observed in the other half of the discussed A_C haplotype, with namely rs34481144:T being rare in Chinese populations (1–2%), rare or infrequent in African groups (2–14%), but fairly frequent in European groups (38–56%). Rs34481144:T was found to correlate strongly with increased influenza severity in three independent cohorts (Allen et al., 2017). These three independent cohorts, however, did not confirm the link between rs12252:G and increased influenza severity, as was suggested in Chinese cohorts. To add to the controversy of the possible antiviral effects linked to rs12252, a detailed study on 293T cells of the putative truncated variant Δ1–21 that is theorized to result from the rs12252:G mutant, showed increased potential to restrict HIV replication and therefore an advantage compared to the complete IFITM3 protein carrying the reference allele (Compton et al., 2016). However, this truncated version was not observed later in the blood of IAV or HIV patients (Randolph et al., 2017; Makvandi-Nejad et al., 2018), while rs12252:G was, on the contrary, found to enhance HIV-1 infection in Chinese patients (Zhang et al., 2015). The reports for the functional role and consequences of minor rs12252:G allele are therefore conflicting and thus inconclusive. Although it is shown that Δ1–21 variant redistributes the protein to the plasma membrane, by prohibiting the phosphorylation of residue Y20 that produces a signal for endocytosis (Jia et al., 2012), a functional link between Δ1–21 and rs12252:G has yet to be established. In the other examined SNP, the minor rs34481144:T allele is currently believed to favor the binding of transcriptional repressor CTCF, also known as CCCTC-binding factor, at IFITM3 promoter, seemingly leading to an inactive IFITM3 profile (Allen et al., 2017). However, the exact functional effect of rs34481144:T is still not well understood.

As part of the IFITM family of proteins, one of the evolutionary ancient first lines of antiviral cellular defenses, the localization in endosomal or lysosomal membrane, or at the

surface, for example, of CD4+ T cells, and the exact antiviral mechanism of IFITM3, is regulated by many different post translational modifications, mainly palmitoylation, ubiquitination and phosphorylation. It is shown that genotypic variants of IFITM3 play a role in diversifying a host's potential antiviral repertoire, in conjunction with selective post translational modifications, and therefore should not be considered de facto as risk factors but rather as trade-offs in antiviral specificity (*Compton et al., 2016*).

This is further supported by the pronounced variability of rs12252:A and rs34481144:C frequencies, which is seen between the major groups in 1000 Genome Project populations, but not as much between subgroups of the same major group. Indeed, the observed spectrum of h1 haplotype prevalence across ancestral populations seems consistent with an evolutionary adaptation to specific immunological challenges and local factors of environmental pressure. In the case of SARS-CoV-2, the observed strong correlation of reference haplotype H1 (A_C) with increased morbidity in ethnic groups in England, could be pointing at a specific antiviral advantage conferred by the presence of each minor allele. However, since both minor alleles are not observed simultaneously (haplotype G_T is not represented), it is harder to conceive an independently equivalent beneficial effect by each distinct minor allele in the mixed reference/minor haplotypes H2 (G_C, here minor allele > rs12252:G) and H3 (A_T, here minor allele > rs34481144:T). Instead, it is more plausible to consider an effective hijacking of IFITM3 by SARS-CoV-2 in order to infect the cell, or to replicate, or to spread, or involving more than one of these phases. Indeed, there are known examples of similar hijacking, for example by the coronavirus that causes the common cold, HCoV-OC43 (*Zhao et al., 2014*), or by human cytomegalovirus (HCMV) (*Xie et al., 2015*). More specifically for HCoV-OC43, it was shown that all three types of interferons, IFN-α, IFN-γ, and IFN-λ, actually enhance HCoV-OC43 infection, while IFITM3 possibly promotes the low-pH–activated membrane fusion between the viral envelope and endosomal membranes. In contrast, human cytomegalovirus hijacks BST-2/tetherin to promote its entry into host cells and co-opts viperin to facilitate its replication, with IFITM3 facilitating the formation of the virion assembly compartment, but the virus is otherwise less sensitive to IFNs.

In the case of SARS-CoV-2, it is therefore not inconceivable that if there is in fact a pro-infection role of IFITM3, that the virus could have evolved to exploit the most abundant haplotype A_C (59% abundance across all populations). The different effect between the H1 haplotype and H2/H3 haplotypes most probably involves the cellular distribution of IFITM3, which is mainly controlled by post-translational modifications, which in turn may be influenced by key polymorphisms such as the two examined here. Of great relevance, in this context, is the finding that plasma membrane localization of IFITM3 enhances SARS-CoV-2 infection, while endocytosis of IFITM3 effectively restricts the virus (*Shi et al., 2020*). The same study confirms an even greater enhancement of SARS-CoV-2 in bypassing IFITM3 defense via TMPRSS2 activation of plasma membrane fusion, and reports compatibility with HCoV-OC43 mode of enhanced infection. Moreover, in another study by *Bozzo et al. (2020)*, IFITM3, together with IFITM2, were shown to boost SARS-CoV-2 infection, rather than restrict it, both in the absence and

presence of interferon, which is consistent with our current suggestion of viral hijacking of IFITM3.

The recent suggestion that rs12252:G is the risk allele in a $n = 80$ COVID-19 cohort with Chinese patients (*Zhang et al., 2020*), appears to challenge our conclusions, claiming the inverse effect. Considering that the cohort took place at Beijing You'an Hospital, if it is safe to assume that patients belonged to EAS-CHB group (Han Chinese in Beijing), the subgroup with the highest frequency in rs12252:G (54%), then an alternative interpretation of the result may be possible. With 28/80 patients hospitalized with pneumonia being homozygotes rs12252:GG, 37/80 being heterozygotes rs12252:AG and 15/80 being homozygotes rs12252:AA, this results to 58% (93/160) abundance for the G allele (minor) and 42% (67/160) abundance for the A allele (reference). As the prior probability for the G allele was as high as 54%, the above result appears inconclusive (i.e., 58% observed vs 54% expected for rs12252:G) and therefore the suggestion that rs12252:G alone is a COVID-19 severity risk allele seems unfounded in this case. The same conclusion is reached, with whichever possible mix of the 3 available Chinese subgroups from 1000 Genomes Projects (EAS-CHB, EAS-CHS, EAS-CDX), as they all show high rs12252:G frequencies (0.47–0.54), surpassed only by the Japanese subgroup (EAS-JPT, 0.64). In the case where heterozygote rs12252:AG (37/80 or 46% abundance) is expected to behave similarly to homozygote rs12252:AA (19%), so that homozygote rs12252:GG (35%) appears as the risk genotype, then a chi-square statistic would report a non-significant $p = 0.36$, when $q$ (frequency of rs12252:G) = 0.54 and $q^2 = 0.29$ according to Hardy-Weinberg equilibrium, or borderline significant $p = 0.04$, when $q$ (frequency of rs12252:G) = 0.47, with $q^2 = 0.22$. Therefore, it is equally unsafe to associate homozygote rs12252:GG with COVID-19 severity.

It is acknowledged that the conclusions of the present investigation take into consideration only allele frequencies and not direct genotyped data. Nevertheless, allele frequencies from large sequencing studies, such as the 1000 Genomes project, are deemed important to be taken into account when available, in order to provide further insight and direction to studies based exclusively on genotypic analysis. In further support to the above, an independent study that compared worldwide COVID-19 mortality statistics with rs12252 allele frequency information per country (extracted from the PubMed database), found that rs12252:G was negatively correlated with the SARS-CoV-2 mortality rate ($p = 0.0008$), in agreement with our current outcomes (*Pati et al., 2020*).

## CONCLUSION

The role of the examined IFITM3 variants in the severity of COVID-19 should be elucidated, by in vitro investigation of the effect of H1 (A_C), vs H2 (G_C), vs H3 (A_T) rs12252_rs34481144 haplotypes in SARS-CoV-2 infectivity and viral spread. Related in silico investigations should include the examined combined haplotype in their analysis and consider non-additive interactions. It is notable how various ongoing GWAS studies (*The COVID-19 Host Genetics Initiative, 2020*; *23andMe, 2020*) did not confirm in their meta-analysis some other independently established genetic variant effects, which are otherwise broadly accepted by the scientific community as influencing COVID-19 severity,

such as the ABO blood group (*Li et al., 2020*; *Wu et al., 2020*), or APOE e4 genotype (*Kuo et al., 2020*). If functional differences between the examined IFITM3 haplotypes are shown to produce distinct profiles of COVID-19 progression in severe patients, then an improved understanding of the underlying mechanisms may allow more adequate or personalized treatment protocols. Last but not least important, is the need for investigating the examined variant implications in raising an effective immune response after future vaccination against SARS-CoV-2, as it was recently demonstrated that homozygote rs12252:GG, actually reduces the level of antibody response after influenza vaccination (*Lei et al., 2020*). It could be important to verify whether this also stands for the upcoming SARS-CoV-2 vaccines, and whether a reduced antibody response could be instead elicited in this case by the major allele (rs12252:A), as was suggested throughout our analysis regarding COVID-19 severity. In conclusion, this study (a) presented one more piece of evidence associating IFTM3 variants with the severity of COVID-19, (b) suggested that the two most highly studied IFITM3 polymorphisms should be considered as a combined haplotype, and (c) is calling for further research focus on this important first line of cellular antiviral defense.

### Funding
The authors received no funding for this work.

### Competing Interests
All authors are employed by Bioiatriki Healthcare Group.

### Author Contributions
- Dimitris Nikoloudis conceived and designed the experiments, performed the experiments, analyzed the data, prepared figures and/or tables, authored or reviewed drafts of the paper, and approved the final draft.
- Dimitrios Kountouras analyzed the data, authored or reviewed drafts of the paper, and approved the final draft.
- Asimina Hiona analyzed the data, authored or reviewed drafts of the paper, and approved the final draft.

### Data Availability
The Standardized Mortality Ratios (SMR) of ethnic groups in England, adjusted for age and region, were adopted from the study by Aldridge et al. in the exact form in which they were presented (*Aldridge et al., 2020*). Specific dataset details, including age, region and ethnicity information, are available at UCL: Aldridge, R; (2020) Dataset: Black, Asian and Minority Ethnic groups in England are at increased risk of death from COVID-19. [Dataset]. UCL Institute of Health Informatics: London, UK. DOI 10.14324/000.ds.10096589.

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
