# Peer review of "The frequency of combined IFITM3 haplotype involving the reference alleles of both rs12252 and rs34481144 is in line with COVID-19 standardized mortality ratio of ethnic groups in England"

_PeerJ, doi:10.7717/peerj.10402_

## Round 0.1 · original submission · Major Revisions

Dear Dr. Nikoloudis and colleagues:

Thanks for submitting your manuscript to PeerJ. I have now received two independent reviews of your work, and as you will see, one reviewer raised serious concerns about the research and recommended rejection. The other reviewer was only a bit more enthusiastic, raising some serious concerns. I agree with the concerns of both reviewers.

Fortunately, there is a lot of criticism here for you to consider. If you choose to address these concerns, I encourage you to revise your work and resubmit. However, I strongly encourage you to take into account all of the concerns raised by both reviewers.

Thanks again for submitting your work to PeerJ.

Good luck with your revision,

-joe

·

Basic reporting

Dimitris N. et al. collected allele frequencies from 1000 Genomes Project to calculate the correspondences with ethnic groups in England. A significant correlation was observed between the reported Standardized Mortality Ratios and the frequency of the combined haplotype of both reference alleles, which suggest that the combination of reference alleles of the specific SNPs may be implicated in more severe outcomes of COVID-19. This field is very important for early targeted intervention in those at high risk.

Experimental design

Research question was well defined, while the idea of creating correspondences with ethic groups and the reported standardized mortality ratios was unreliable.

Validity of the findings

The findings and results are quite uncertain, especially there is no precise basic information to support this analysis, for example, age, gender, health status, medical conditions, which were important factors to outcome of COVID-19. Please provide solid evidence to support the study.

Additional comments

The research question is very important and further analysis with more precise data is necessary to support the findings.

·

Basic reporting

The manuscript is written in clear, understandable English however I suggest proof reading of the abstract and introduction for grammatical errors. Specifically:
- Line 18-19: "Following up to these two developments..." should read "Following up on these two developments..."
- Line 65: "...turned our focus into..." should read "turned our focus to..."

The literature review is well written and clearly introduces all of the necessary information. I would suggest including previously described data showing IFITM protein restriction of SARS-CoV (Huang et al, Plos Path, 2011).

I would suggest combining the results into one figure with 3 panels as they are repetitive and some are not necessary or unclear. I also suggest finding a way to show the SMR and h1 ratio with the same ethnic categories to show that they share the same pattern on Figure 1. You could add SMR with a second y-axis for example. Figure 2a is unnecessary as it is the same as Figure 1. Figure 3 does not add anything as the correlation is much more convincing in figure 4. I suggest having one figure showing (a) ratios, (b) SMR and (c) correlation between h1 ratio and SMR.

I would suggest adding a leading sentence to the results section explaining what you have done briefly.

Experimental design

The design is robust and the authors clearly explain the limitations of their ethnic pools and groupings.

Validity of the findings

The findings seem valid, and the authors discuss how these findings link to previous data well.

Additional comments

The manuscript is well written and presents and interesting observation in regards to the link between COVID-19 mortality and IFITM3.

---

## Round 0.2 · Minor Revisions

Dear Dr. Nikoloudis and colleagues:

Thanks for revising your manuscript. The reviewer is mostly satisfied with your revision (as am I). Great! However, there are a few remaining concerns to address. The figure legends need to be reworked (with more accurate descriptions). The Discussion should address observed allelic frequencies.

Please address these ASAP so we may move towards acceptance of your work.

Best,

-joe

·

Basic reporting

Grammatical errors previously highlighted have been amended.

The figures have improved slightly but I feel the figure legends do not describe the data well enough and instead try to present results, including those that are not even shown in the figure. Better description of the figures is required.

Experimental design

Previous comments have been responded to.

Validity of the findings

I feel the discussion would benefit from the acknowledgement that this data only takes into consideration minor allele frequencies and not genotype data. The association between rs12252 and rs34481144 and viral infection severity is generally linked to genotype not minor allele frequency, with heterozygosity for each SNP generally behaving more similarly to the wild-type homozygous genotype.

---

## Round 0.3 · accepted · Accept

Dear Dr. Nikoloudis and colleagues:

Thanks for revising your manuscript based on the concerns raised by the reviewer. I now believe that your manuscript is suitable for publication. Congratulations! I look forward to seeing this work in print, and I anticipate it being an important resource for groups studying the role of IFITM3 variants in the mechanism of cellular invasion of SARS-CoV-2. Thanks again for choosing PeerJ to publish such important work.

Best,

-joe